# Evaluation of Customs Supervision Competitiveness Using Principal Component Analysis

**Rong Hu** [1,*], **Yui-Yip Lau** [2,*] and **Ruiqian Wang** [3]

1 School of Customs and Public Administration, Shanghai Customs College, Shanghai 201204, China
2 Division of Business and Hospitality Management, College of Professional and Continuing Education, The Hong Kong Polytechnic University, Hong Kong, China
3 Dalian Customs District, China Customs, Dalian 116001, China
* Correspondence: hurong@shcc.edu.cn (R.H.); yuiyip.lau@cpce-polyu.edu.hk (Y.-Y.L.)

**Abstract:** In order to improve the degree of security and facilitation of the business environment; customs administrations are constantly working to strengthen their own institutional innovation and governance in customs control. As such, this paper establishes an evaluation index of international customs supervision competitiveness based on the eight indexes extracted from the World Customs Organisation (WCO) Revised Kyoto Convention and selects 21 representative national customs using the principal component analysis (PCA) method to assess their competitiveness against SPSSAU quantitatively. Based on the data from the World Economic Forum, World Bank, OECD, WCO Annual Report, and Transparency International, the Dutch customs have relatively the best performance in the range of comprehensive competitiveness, and customs authorities in Germany, New Zealand, the United Kingdom, the United States, Mexico, Australia, the Netherlands, and Singapore also have relatively-best performance under different indexes. Taking China Customs as an example, the gaps between China Customs and the ones with the best performance are also analyzed. In response to the problems identified by the analysis, recommendations are made in the areas of process facilitation, technology application, international cooperation, economic development, taxation management, and capacity building to improve the competitiveness of customs control.

**Keywords:** customs administration; international customs supervision competitiveness; World Customs Organization Revised Kyoto Convention; customs control





## 1. Introduction

From the point of World Customs Organisation (WCO) practice, the WCO has played a leading role in the area of organizational performance measurement in recent years. Veenstra and Heijmann (2022) [1] addressed that customs played a key role in due diligence in global supply chains. To a certain extent, customs supervision fosters resilient and sustainable transport and trade facilitation. Specifically, customs may play a key role in exhibiting spot checks at the border, gathering information on the correctness of reporting by the different firms, and seizing products that are an obvious infringement of the regulation. Customs supervision performs a determined role in controlling compliance with standards associated with the UN 2030 Agenda goals in the discipline of global trade in commodities. In 2015, WCO created the preliminary model for performance measurement with four primary indicators, 14 secondary indicators, and 20 tertiary indicators. Later, the Working Group on Performance Measurement (WGPM) was established to develop a WCO performance measurement mechanism (PMM) for comprehensive performance measurement that covered all customs competencies. PMM would be recognized as the leading standard for performance measurement in customs areas. With the already-achieved milestones, such as the scope and criteria of key performance indicators (KPIs), the PMM dimensions, and their expected outcomes against the sustainable development goals (SDGs). WGPM

still needs to work further on the set and sub-set of KPIs for the common dimensions of performance measurement, the disclosure policy, and the assessment circle [2]. In 2021, WGPM successfully concluded its 3rd and 4th meetings. During the 3rd meeting, the WGPM reached a common understanding on the list of expected outcomes for the approved dimensions of performance under the PMM and acknowledged the first draft set of KPIs, thereby making further observations that demonstrated delegates' keen interest in continuing the discussions to ensure their reliability and global relevance [3]. During the 4th meeting, key milestones, such as the specifications for the hybrid model (e.g., voluntary self-assessment followed by a peer review stage, at members' request, to ensure a thorough performance evaluation) and the decision to move progressively towards the development of a WCO data collection platform to ensure adequate analytical capabilities in support of the mechanism and its usability, were reached on several aspects [4].

From the point of WCO research, performance measurement has been discussed at the international grand level and also in the context of a single country. At the international level, the paper concerning an overview of performance measurement in customs administration considers four broad approaches, namely, customs data mining, service charters, perception indexes, and monitoring mechanisms. The paper concludes that performance measurement should primarily be about improving the effectiveness and efficiency of customs administration functions [5]. By discussing the relationship between business environment and customs, another WCO research conducted a comparative study on cross-border trade indicators. This research mainly compared the differences among the ease of doing business index, logistics performance index, and trade facilitation index and emphasized the importance of the WCO time release study [6]. At the level of a single country, another paper that focused on Korea has examined the performance of the Korea Customs Service (KCS) selectivity system, drawing on practices used in the fields of taxation and insurance that deal with similar kinds of risks or frauds. The KCS currently uses three selection methods, namely, manual, rule-based, and random selection. The paper analyzes these results and concludes that the three selection methods are complementary for the detection and deterrence of emerging and evolving risks [7].

The WCO has made constant progress in exploiting the system and methods for measuring customs performance, and its existing achievements and ideas have been absorbed widely in related research about national customs performance measurement and the international comparison of customs performance measurement amongst different countries. Studies on the competitiveness of customs supervision are constantly emerging with the modernization of the national governance capacity in China. At present, domestic and foreign existing studies focus more on the proposal of objectives and the setting of evaluation systems and indexes at the macro level, whilst quantitative research at the micro level is relatively rare. For the existing quantitative research, most of them are limited to the internal comparison of China Customs, and the comparative study of international customs competitiveness is rare. On the one hand, a more general evaluation system with indexes for international comparisons must be built; on the other hand, the use of standardized, and unified sources of data are also the premise of scientific research on this issue. The data and statistical methods of comparative research on international competitiveness are in the exploration stage, which is the breakthrough point of this paper.

Customs performance is important, and we need a scientific and efficient tool to measure it to fulfill the goal. We had to create a trustful measurement system and use a reliable methodology to rank the performance. Finally, we need to determine the core elements of the performance measurement system. Based on the research results, we can provide suggestions on customs administrations to improve their performance in the following days.

## 2. Literature Review

From the point of other research made by scholars or customs officials from different countries other than China, this study can be divided mainly into four parts, namely, the

measurement of customs performance through various quantitative methodologies, the tools or technologies suitable for or beneficial to customs performance management, the outcome of performance management for national customs or their sub-divisions and the application of performance management in different customs operations.

The first part is about quantitative methodology, principal component analysis (PCA), data envelopment analysis (DEA), and the combination of panel data regression and growth curve analysis have been used in terms of the evaluation of the organizational performance of different customs services. PCA is a useful statistical tool for the source apportionment of trace elements in PM10 of environment protections [8]. It is also widely used in the spatial assessment of water quality parameters [9] and the early detection of process faults in fault detection technologies [10]. PCA reveals that three PCs (e.g., drugs and substance abuse, unemployment, and neglect from parents) explain approximately 52.6% of the total variability of the causes of crimes against the person and are suggested to be retained [11]. This research indicated that PCA could be used not only in the rank tools but also in finding root causes.

Based on data from the WTO, WCO, and WB, PCA has been used to assess the competitiveness of different customs authorities from 29 countries under the international trade framework. A total of five principal components have been finally extracted from 17 indicators. In descending order, the result shows that the most competitive customs authorities assessed include Panama, followed by China, India, Germany, Korea, Sweden, Singapore, Turkey, Thailand, and Chile [12]. During the 15th WCO PICARD Conference in 2020, organizational performance measurement was among the heated topics, with presenters using different methods for scientific evaluation. DEA has been used to assess the relative efficiency of the implementation of WCO Policies and Guidelines by the customs and commercial community on the basis of two models, namely, constant return to scale (CRS) and variable return to scale (VRS). By also collecting data from the WTO, WCO, and WB from three periods, this study comes to a conclusion that: Over the years, the countries have had a declining interest in the effective implementation of the policies and guidelines issued by the WCO, improving the levels of policy implementation and guidelines generated by the WCO is necessary [13]. In order to investigate the impact of the implementation of balanced scorecards (BSC) as a performance management system on organizational performance in the Indonesian customs and excise administration, the panel data regression and growth curve analysis has been used with the outcomes showing that the use of BSC as the performance management system has a positive association with organizational performance, thereby suggesting that the usage of BSC in Indonesian public sector organizations has empirical support thereafter [14].

The second part is about suitable tools or technologies to improve customs performance. Looking at how the uptake of artificial intelligence (AI) has affected customs administration in terms of transit management and security of cargo, as well as trade facilitation, which greatly reduces the cost of doing business by the private sector. The study from Uganda Revenue Authority offers several recommendations for policymakers, including undertaking private-public partnerships (PPP), integration of RECTS with other customs systems, well-planned change management, and developing a pool of AI experts in customs [15]. Another research conducted by an Italian expert shows that modern customs should use data collection and analysis techniques to facilitate trade not only by minimizing obstacles for operators in terms of the fluidity of their operations but by observing and analyzing their behavioral patterns to introduce simplifications in customs procedures aiming to make them more user-friendly [16]. Back in 2012, a work conducted by WCO technical officer provided an introduction to the Time Release Study (TRS) Guide Version 2 developed by the WCO in 2011. It includes an overview and new aspects of the WCO TRS guide and examples of TRS results. This paper also covers the main focus of the TRS and explores ways of using the TRS methodology in an international environment to measure the performance of a supply chain and an international corridor, which are key to strengthening regional cooperation and integration further [17].

The last two parts are about the performance measurement of national customs and specific areas of customs operations. Two types of research analyzes the performance of Russian customs authorities. The first one finds out that the existing performance indicators do not form part of a single system; they are often duplicated and are differentiated by customs authorities' levels or tiers. Then, it highlights a number of difficulties during the calculation of the indicators and inconsistencies in their application and documents. The first research proposes an alternative, dual-purpose system of indicators that have been constructed from the perspectives of the participants of international economic activities and customs authorities. The results of the study should serve as a basis for the creation of a more practical model for assessing the performance of customs authorities [18]. The second paper examines customs performance measures and various indicators in light of the drawbacks and limitations of the current system of performance measurement adopted by the Russian Federal Customs Service. The existing system of customs performance measurement is far from perfect and, to some extent, can cause problems in itself. This paper explains how performance can be measured in a way that improves the current system by ensuring it achieves the objectives of customs service more comprehensively and enhances its overall effectiveness and efficiency [19]. Finally, the performance measurement of customs in AEO has been analyzed. The study compares the analysis of the indicators found in the literature to the indicators at the regulatory level and provides a useful opportunity to unveil the AEO indicators in an implementing country [20].

From the point of other research made by scholars or customs officials from China, many Chinese experts and scholars have conducted research on the most competitive customs supervision mechanism. China customs aims at the goal of building the most competitive customs supervision system in the world [21]. The specific realization routine relies on 'five customs construction' as the core, with a system of index formulated for political leadership, customs service, working efficiency, technological support, regulations, integrity, and comprehensive security [22]. The internationally competitive customs supervision mechanism based on the index system of trade security and facilitation, system construction and the development of human resources, fair and efficient taxation, and law enforcement should be further studied [23]. Factor analysis is used to evaluate the competitiveness of every customs office in a regional custom from the aspects of customs declaration volume, tax collection, tax deduction exemption, processing record funds, and tax reimbursement for domestic sales [24]. An index system for the evaluation of customs competitiveness through international comparison and reference has been built [25].

In addition, some scholars have analyzed and compared the competitiveness in the field of international trade. For example, the backpropagation algorithm has been used to evaluate seven customs offices from the aspects of electronic customs clearance, inspection equipment, customs clearance mode, clearance time-consumption, organization learning, and the core competitiveness of third-party enterprises [26]. The competitiveness of service trade has been compared from the international market share, special trade coefficient (TC), revealed comparative advantage index (RCA), and service trade openness index (STO) [27].

## 3. Materials and Methods

### 3.1. Evaluation Index Elements of International Competitiveness of Customs Supervision and Data Sources

The Revised Kyoto Convention is the only international legal document that comprehensively regulates the standards of customs systems and practices in the world today. It is an important basis for the WCO to promote the coordinated development of customs systems and practices in various countries and regions around the world and is also an important reference standard for each country and region to formulate its own customs systems to strengthen supervision and promote and facilitate the development of trade. Judging from the revision work of the Revised Kyoto Convention launched in 2019, the customs supervision of all member countries still follows the aforementioned framework.

Therefore, the evaluation index system of international competitiveness for customs from nearly all countries in the world can almost fully refer to the Revised Kyoto Convention.

According to the Revised Kyoto Convention, technical standards are mainly set up in the areas of customs clearance procedures, duties, guarantee (security), customs supervision, information and communication technology, customs relationship with third parties, information, and decisions and rulings provided by customs and customs affairs complaints and appeals. This provides an ideal basis for the evaluation of customs competitiveness internationally.

Based on the eight aforementioned, the paper selects suitable data for analysis from the data set of the World Economic Forum, World Bank, the trade facilitation database of OECD, WCO annual reports, and Transparency International. For the customs clearance procedure, the paper uses 'burden of customs procedures' from the World Economic Forum, 'logistics performance' from World Bank, and 'average import and export time' from the World Bank to measure the first index. For the duties, the paper uses 'tax revenue' from the World Bank and 'fee and charges' from OECD to measure the second index. For the guarantee (security), the paper uses 'WCO instruments' from WCO annual reports to measure the third index. For customs supervision, the paper uses 'customs officers' productiveness' from WCO annual reports to measure the fourth index. For information and communication technology, the paper uses 'automation' from OECD to measure the fifth index. For the customs relationship with third parties, the paper uses 'involvement of trade community,' 'external border agency cooperation,' and 'internal border agency cooperation' from OECD to measure the sixth index. For the information & decisions, and rulings, the paper uses 'information availability' and 'advance ruling' from OECD to measure the seventh index. For the customs affairs complaints and appeals, the paper uses the 'Corruption Perceptions Index' (CPI) value from Transparency International and the 'appeal procedure' from OECD to measure the eighth index.

### 3.2. Data and Sample Country Specification

As mentioned above, the data collected for the analysis in this paper originate from the statistics published by international organizations, such as the WCO, the World Bank, the World Economic Forum, OECD (Organisation for Economic Co-operation and Development), and Transparency International. Therefore, the data collection is authoritative and objective.

#### 3.2.1. Data Specification

Three points need to be noted about the used data. The first is timeliness. Due to the lag of statistics and release, the most data that can be collected at present is for 2019, which is still of practical significance for establishing the international competitiveness of customs supervision of various countries. The World Bank releases logistics performance data, which measure regulatory effectiveness, every two years because no data are available for 2019, the data for 2020 have not been published yet, and the data for 2018 are used for the evaluation. The second is the transparency of customs. Considering that the transparency of a specific institution, such as customs, is seldom published internationally at present, this paper replaces the transparency of customs with data on the transparency of the countries. The third point is about data pre-treatment, including unified and standardized dimensionless processing and partial index reverse pre-processing (see Table A1).

#### 3.2.2. Sample Country Specification

This paper adopts the data from Group of Twenty, including China, Argentina, Australia, Brazil, Canada, France, Germany, India, Indonesia, Italy, Japan, South Korea, Mexico, Russia, Saudi Arabia, South Africa, Turkey, the United Kingdom, the United States, and the European Union The use of G20 is of typical significance to evaluate the international competitiveness of customs supervision: firstly, G20 is the most vigorous economic organization in today's global economy, and G20 itself contains developed and developing countries

with wide typicality. Meanwhile, on the basis of G20, the Netherlands, New Zealand, Singapore, and other national customs services with global bench-marking significance in customs supervision have also been included in the analysis. Due to the lack of relevant data on Russia in the World Bank and the WCO database, and the EU is not a single country, these two subjects are excluded. The final countries involved in the evaluation are 21 in total, and the samples can sufficiently reflect the international competitiveness of customs supervision (see Table A2).

### 3.3. Principal Component Analysis

3.3.1. Brief Introduction to the Method

Principal component analysis (PCA) is a widely used data dimension reduction algorithm. The main idea of PCA is to map the features of the n-dimension to the k-dimension, which is a new orthogonal characteristic, also known as the principal component, and forms new k-dimension features based on the original n-dimension ones. PCA is used to find a set of coordinate axis orthogonal to each other from the original space. The selection of a new axis is closely related to the data itself. The first newly selected axis is the direction with the largest variance in the original data, the second new axis is in the plane, which is orthogonal to the first axis, to obtain the largest variance, and the selection of the third axis is in the plane orthogonal to the first and second new axes to make the largest variance. This process is known as varimax.

As an analogy, we can obtain n axes. In this way, most of the variances are contained in the first k axes, and the variances in the later axes are almost zero. Therefore, we can ignore the rest of the axes and only keep the first k axes with most of the variance. In fact, this is equivalent to only retaining the features from dimensions that contain most of the variance and ignoring the features from dimensions that contain almost zero variance to reduce the feature dimension of the data. In this paper, the eigenvalue decomposition of the covariance matrix is used for PCA.

1. A standardized collection of raw index data: p-dimensional random vector x = ($X_1$, $X_2$, ..., $X_p$)T with n samples $x_i$ = ($x_{i1}$, $x_{i2}$, ..., xip)T, i = 1, 2, ..., n, n > p, the sample matrix is created, and the sample matrix elements are standardized as follows:

$$Z_i = \frac{x_i - x_j}{s_j}, i = 1, 2, \ldots, n; j = 1, 2, \ldots, p$$

$$\overline{x}_j = \sum_{i=1}^{n} x_{ij}, s_j^2 = \frac{\sum_{i=1}^{n} (x_{ij} - \overline{x}_j)^2}{n-1}$$

The standard matrix Z is obtained

2. Find the correlation coefficient matrix of the standard matrix Z.

$$R = [r_{ij}]_p xp = \frac{Z^T Z}{n-1}$$
$$r_{ij} = \frac{\sum z_{kj} \cdot z_{kj}}{n-1}, i, j = 1, 2, \ldots, p$$

3. Solve the characteristic equation of sample correlation matrix R.

$$\lfloor R - \lambda I_P \rfloor = 0$$

P characteristic roots are obtained, and principal components are determined.

m is measured through $\frac{\sum_{j=1}^{m} \lambda_j}{\sum_{j=1}^{p} \lambda_j} \geq 0.85$, thereby making the utilization rate of information reach more than 85%, for each λj, j = 1, 2, ..., m, solve the equations Rb = λjb, and the unit eigenvector $b_j^0$ is obtained.

4. Transform the standardized index variables into principal components.

$$U_i = z_i^T b_j^0, j = 1, 2, \ldots, m$$

U1 is the first principal component, U2 is the second principal component, and Up is the p principal component.

5.　　Evaluate m principal components comprehensively.

The final evaluated value is obtained by the weighted summation of m principal components, and the weight is the variance contribution rate of each principal component.

### 3.3.2. Analysis of Software

In this paper, SPSSAU is selected to conduct PCA and calculations. The functions include information condensation, which means multiple analysis items are condensed into several key general indexes; weight calculation to use the value of variance interpretation rates to calculate the weight of each general index; comprehensive competitiveness: using the two indexes of component score and variance interpretation rate, and the comprehensive score can be calculated and used for the comparison of comprehensive competitiveness (the higher the comprehensive score is, the stronger the competitiveness is). This paper mainly uses PCA to compare the comprehensive competitiveness across different national customs.

### 3.3.3. Applicability Test

PCA automatically generates the weight of each principal component through sample data, which largely resists the interference of human factors in the evaluation process. Meanwhile, an applicability test will be conducted to prove that the principal component of comprehensive evaluation theory provides a scientific and objective evaluation method.

Before using this method for information condensation research, we first analyze whether the research data are suitable or not for PCA. Through calculations, KMO is 0.686, which meets the prerequisite that the PCA can be used if KMO is greater than 0.6 (Table 1). The data also passed the standard of the Bartlett Sphericity Test ($p < 0.05$), indicating that the sample data are very suitable for PCA. Table 2 presents the weight result of the linear combination coefficient of PCA.

**Table 1.** Applicability of PCA.

| KMO | | 0.686 |
|---|---|---|
| | c2 | 196.71 |
| Bartlett Sphericity Test | df | 105 |
| | p | 0 |

**Table 2.** Variance Interpretation Rate.

| Item | Characteristic Root | | | Principal Component | | |
|---|---|---|---|---|---|---|
| | Characteristic Root | Variance Interpretation Rate% | Accumulation% | Characteristic Root | Variance Interpretation Rate% | Accumulation% |
| 1 | 7.116 | 47.441 | 47.441 | 7.116 | 47.441 | 47.441 |
| 2 | 1.604 | 10.695 | 58.136 | 1.604 | 10.695 | 58.136 |
| 3 | 1.581 | 10.541 | 68.677 | 1.581 | 10.541 | 68.677 |
| 4 | 1.037 | 6.915 | 75.592 | 1.037 | 6.915 | 75.592 |
| 5 | 0.956 | 6.374 | 81.966 | - | - | - |
| 6 | 0.608 | 4.056 | 86.022 | - | - | - |
| 7 | 0.514 | 3.429 | 89.451 | - | - | - |
| 8 | 0.409 | 2.726 | 92.177 | - | - | - |
| 9 | 0.345 | 2.298 | 94.475 | - | - | - |
| 10 | 0.289 | 1.928 | 96.403 | - | - | - |

**Table 2.** *Cont.*

| Item | Characteristic Root | | | Principal Component | | |
|------|---------------------|---|---|---------------------|---|---|
| | Characteristic Root | Variance Interpretation Rate% | Accumulation% | Characteristic Root | Variance Interpretation Rate% | Accumulation% |
| 11 | 0.218 | 1.456 | 97.859 | - | - | - |
| 12 | 0.133 | 0.885 | 98.744 | - | - | - |
| 13 | 0.110 | 0.733 | 99.477 | - | - | - |
| 14 | 0.055 | 0.368 | 99.845 | - | - | - |
| 15 | 0.023 | 0.155 | 100.000 | - | - | - |

Table 3 shows that the corresponding commonality value (common factor variance) of all research items is higher than 0.4, which means a strong correlation between research items and principal components. The principal components, in this case, can effectively extract information. After ensuring that the principal components can extract most of the information of the research items, then this paper analyzes the corresponding relationship between the principal components and the research items (when the absolute value of the load coefficient is greater than 0.4, the items have a corresponding relationship with the principal components).

**Table 3.** Loading Factor Table.

| Name | Loading Factor | | | | Common Factor Variance |
|------|----------------|---|---|---|------------------------|
| | Componet1 | C2 | C3 | C4 | |
| Customs clearance and procedures 1 | 0.856 | −0.071 | 0.037 | −0.249 | 0.801 |
| Customs clearance and procedures 2 | 0.934 | 0.116 | 0.151 | −0.057 | 0.912 |
| Customs clearance and procedures 3 | 0.714 | −0.401 | −0.253 | 0.030 | 0.736 |
| Duties-1 | 0.674 | −0.154 | −0.092 | −0.442 | 0.682 |
| Duties-2 | 0.663 | 0.302 | −0.313 | 0.491 | 0.870 |
| Guarantee Security | 0.143 | −0.044 | 0.872 | 0.104 | 0.793 |
| Customs Control | 0.503 | 0.285 | −0.463 | −0.317 | 0.649 |
| Information and communication technology | 0.714 | −0.283 | −0.346 | 0.264 | 0.779 |
| Customs with 3rd parties-1 | 0.798 | −0.018 | −0.033 | −0.250 | 0.700 |
| Customs with 3rd parties-2 | 0.647 | −0.228 | 0.176 | 0.411 | 0.670 |
| Customs with 3rd parties-3 | 0.550 | −0.656 | 0.138 | 0.020 | 0.752 |
| Information, decision, and ruling-1 | 0.787 | −0.057 | 0.294 | 0.001 | 0.709 |
| Information, decision, and ruling-2 | 0.636 | 0.447 | −0.131 | 0.323 | 0.725 |
| Customs affairs complaints and appeals-1 | 0.826 | 0.189 | 0.193 | −0.062 | 0.759 |
| Customs affairs complaints and appeals-2 | 0.513 | 0.652 | 0.316 | −0.116 | 0.802 |

## 4. Results and Discussion

### 4.1. Overall Scores of International Competitiveness Comparison

According to the PCA results, the competitiveness of 21 national customs (Table 4) and some other findings can be determined by ranking the comprehensive scores.

Based on the RKC elements and all indexes from different international organizations, the highest-ranked country is the Netherlands, which means that the Dutch customs is not only prominent in the aspect of trade facilitation but also possesses high-quality customs control and risk management. It aligns with the fact that the Dutch customs is quite famous for their risk management scheme. Generally, the customs performance of developed countries in the area of customs competitiveness is better than those of developing countries. Although New Zealand and Singapore customs are quite strong in the area of facilitation, the evaluation of customs competitiveness contains components that cover customs security and trade facilitation. The ranks of Singapore and New Zealand customs are 7th and 8th, respectively. The table also shows the remarkable performance of

customs authorities in South Africa with strong support in capacity building from the WCO.

**Table 4.** Final Scores.

| Country | CompScore | RANK |
|---|---|---|
| Netherlands | 0.749506453 | 1 |
| Germany | 0.726227359 | 2 |
| USA | 0.717200976 | 3 |
| Japan | 0.592219476 | 4 |
| France | 0.554168705 | 5 |
| Australia | 0.550103516 | 6 |
| Singapore | 0.533268583 | 7 |
| New Zealand | 0.512146023 | 8 |
| Britain | 0.442885783 | 9 |
| Korea | 0.399528613 | 10 |
| Canada | 0.270962187 | 11 |
| China | −0.035276941 | 12 |
| South Africa | −0.036831434 | 13 |
| Italy | −0.060654602 | 14 |
| Saudi Arabia | −0.528623324 | 15 |
| Argentina | −0.65178987 | 16 |
| India | −0.750379569 | 17 |
| Turkey | −0.870228165 | 18 |
| Brazil | −0.913441113 | 19 |
| Indonesia | −1.042272813 | 20 |
| Mexico | −1.158719844 | 21 |

*4.2. Analysis of Customs Competitiveness and the Reasons for the Gaps*

Through the analysis of various components with the comprehensive scores shown in Table 5, we can find what has been achieved and the gaps that remain compared with the highest-standard practice of related national customs. From the top performance customs organizations, we can learn their best practice and improve our performance, which will help us understand the gap and determine the way to improve further. From the weight, we also can learn the key factor in the customs control regime. We can exert our main efforts into the most important area.

**Table 5.** Linear Combination Coefficient and Weight Results.

| Name | PC1 | PC2 | PC3 | PC4 | Comprehensive Score Coefficient | Weight |
|---|---|---|---|---|---|---|
| **Characteristic Root** | **7.116** | **1.604** | **1.581** | **1.037** | | |
| **Variance Interpretation Rate** | **47.44%** | **10.69%** | **10.54%** | **6.92%** | | |
| Customs clearance and procedures 1 | 0.3208 | −0.0562 | 0.0292 | −0.2444 | 0.1751 | 7.22% |
| Customs clearance and procedures 2 | 0.3501 | 0.0919 | 0.1202 | −0.0563 | 0.2443 | 10.07% |
| Customs clearance and procedures 3 | 0.2678 | −0.3169 | −0.2012 | 0.0298 | 0.0979 | 4.04% |
| Duties-1 | 0.2527 | −0.1217 | −0.0730 | −0.4339 | 0.0915 | 3.77% |
| Duties-2 | 0.2484 | 0.2387 | −0.2493 | 0.4824 | 0.1990 | 8.21% |

**Table 5.** *Cont.*

| Name | PC1 | PC2 | PC3 | PC4 | Comprehensive Score Coefficient | Weight |
|---|---|---|---|---|---|---|
| **Characteristic Root** | **7.116** | **1.604** | **1.581** | **1.037** | | |
| **Variance Interpretation Rate** | **47.44%** | **10.69%** | **10.54%** | **6.92%** | | |
| Guarantee security | 0.0536 | −0.0349 | 0.6934 | 0.1022 | 0.1347 | 5.56% |
| Customs control | 0.1885 | 0.2254 | −0.3681 | −0.3115 | 0.0704 | 2.90% |
| Information and communication technology | 0.2676 | −0.2234 | −0.2753 | 0.2595 | 0.1217 | 5.02% |
| Customs with 3rd parties-1 | 0.2990 | −0.0139 | −0.0262 | −0.2454 | 0.1596 | 6.58% |
| Customs with 3rd parties-2 | 0.2424 | −0.1800 | 0.1401 | 0.4032 | 0.1831 | 7.55% |
| Customs with 3rd parties-3 | 0.2062 | −0.5177 | 0.1099 | 0.0200 | 0.0733 | 3.02% |
| Information, decision, and ruling-1 | 0.2949 | −0.0453 | 0.2339 | 0.0012 | 0.2114 | 8.72% |
| Information, decision, and ruling-2 | 0.2383 | 0.3527 | −0.1044 | 0.3173 | 0.2139 | 8.82% |
| Customs affairs complaints and appeals-1 | 0.3096 | 0.1492 | 0.1531 | −0.0605 | 0.2312 | 9.53% |
| Customs affairs complaints and appeals-2 | 0.1924 | 0.5147 | 0.2515 | −0.1140 | 0.2182 | 9.00% |

### 4.2.1. Index Weights over 9% and the Related Top Scoring Customs Administrations

The index that affects the score of comprehensive competitiveness of customs clearance with the highest weight is 'logistics performance,' which accounts for 10.1%.

The logistics performance of German customs indicates that the EU customs union has great achievements in customs control. Germany also plays the role of an important logistics hub in the global supply chain, depicting that logistic performance and customs control can influence each other positively. The index also shows that customs control innovation can have an intensive and deep influence on logistics.

The logistics performance index measures the improvement of the overall logistics performance caused by the simplification of customs clearance procedures. The scores of China customs on the index are tied for the 12th place together with South Korea. On the one hand, this finding reflects that China customs has made remarkable achievements in continuously optimizing the business environment (e.g., the streamlining of attached documents, paperless customs clearance, the national integration of customs clearance, two-step declaration, and the exploration of two-wheel drive), thereby resulting in the improvement of import and export document procedures, simplification of customs procedures and the reduction in customs clearance time. These customs-led reforms have played a continuous role in accelerating the efficiency of customs clearance. On the other hand, we can see the gap between China Customs and German Customs. As the country with the highest score in terms of logistics performance, Germany applies the customs clearance mode of the European Union as a member state. The use of manifest (through Entry Summary Declaration) has been mature, and the risk prevention and control of entry and exit basically depend on the manifest logistics data. Comparatively, the current risk prevention and control of China Customs is gradually transforming to relying on the risk analysis of logistics manifest, and the data quality still needs to be improved further.

The second weight index is the 'CPI value,' which accounts for 9.5%. New Zealand customs achieve ideal performance in terms of integrity. The national anti-corruption index from Transparency International is used to replace the relevant index that reflects customs anti-corruption and complaints, which is the weakness of the PCA analysis. Due to the lack of relevant international data, we can only take alternative values into analysis, which may lower the ranking of China customs in international customs competitiveness as a whole.

The third weight index is 'appeal procedures', which accounts for 9.0%. German customs shows its excellent performance in the areas of judicial appeals, appeal lodging time, appeal delays, appeal information motives, appeals introduced by customs, appeals introduced by traders, administrative appeals number, judicial appeals number, judicial

appeal time limit, appeal time limit decision, legal framework efficiency, and judicial independence according to the dimensions of this index from OECD.

### 4.2.2. Index Weights between 7% and 9% and the Related Top Scoring Customs Administrations

The fourth weight index is 'information availability,' which accounts for 8.8%. The best-performance customs administrations, in this case, are the customs of the UK, the US, and Mexico. The data from OECD measures the following elements: customs website, online feedback, rate of duties information, inquiry points, inquiry points operating hours, inquiry points timeliness, import/export procedure information, accessible documentation, advance publication, advance publication-time, agreements publication, appeal procedures information, customs classification examples, advance rulings information, breaches formalities, application legislation, judicial decision, professional users site, user manuals, website user-friendliness and policy-making transparency.

The fifth index is 'advance ruling,' which accounts for 8.7%. Australian customs is the role model for implementing the advanced ruling system and scheme. As the main measures in the TFA, advance ruling helps customs to improve efficiency and effectiveness.

The sixth index is 'customs fee and charges', which accounts for 8.2%. The customs of the Netherlands, the US, and New Zealand are the ones with the best performance among the 21 customs authorities. The index includes information on fees, fees for evaluation, fees for all-inclusive information, number of fees collected, fees for inquiry, fees for periodic review, fees for advance publication, fees for normal working hours, penalties, penalties for disciplines, penalties for procedural guarantees, penalties for conflicts of interest and penalties for voluntary disclosure. China and Singapore obtain the same performance in this scope, which means that China customs has achieved a lot in the normalization of the collection of fees and charges.

The seventh index is 'external cooperation,' which accounts for 7.6%. British customs play a role model in this area. China customs has noticed the importance of and put more effort into the 'three smarts' construction with more countries. At the same time, China Customs have also noticed its weakness in cooperation with other domestic government agencies and will improve it constantly. This agenda has been included in China Customs' Fourteenth Five-Year Strategic Plan.

The eighth index is 'customs clearance,' which accounts for 7.2%. Singapore is the country with the highest score, essentially because of the nature of its geographic condition as a transit port. Customs in Singapore do not need enterprises to attach regulatory documents to a large number of declarations, which is different from the nature of other countries import and export goods supervision in the hinterland. The second reason is that Singapore's single window has been upgraded from Tradenet to NTP (Networked Trade Platform) to build a national trade information ecosystem and bring traders, logistics service providers, freight forwarders, and banks together on the same platform so that traders can obtain various governmental and commercial services at the same time.

As explained above, the main reason for the result related to the index of 'customs clearance' is due to Singapore's implementation of the single window and simplified procedures on clearance. However, considering the trend to implement single windows across many customs administrations, the gap among countries on clearance continues to be reduced and eliminated, which successfully explains why this index only weighs 7.2%.

### 4.3. Others

Finally, it should be pointed out that although the 'production efficiency index' has a low weight in the overall competitiveness ranking, accounting for only 2.90%. However, the original data on the 'production efficiency' of China Customs officers are that each customs officer processes 835 customs declarations, and the highest is Singapore, which is 10 times that of China, reaching 8573. The top three countries: Singapore (8573), Germany (6738), and South Korea (6520), all have international shipping centers, with a large number

of transit goods under customs supervision through, for example, the Singapore port, Hamburg Port and Busan Port. This depicts that, on the one hand, the policies of China Customs to actively support free trade ports, pilot free trade zones, and international shipping centers are correct and should continue to be adhered to. On the other hand, it also shows that China Customs' regulatory policies still have room to be optimized and can further support the development of the international transit business. Finally, it can be seen that the efficiency of China Customs officers is far behind that of the developed countries.

From the above-mentioned analysis, we can find that logistics performance is the key factor in improving customs control. That is also the reason why USA CBP (USA customs administration), EU customs, and Japanese customs implement the logistics information into customs risk management and put more and more effort into the logistics providers' control. Capacity building, including integrity, is still the bottleneck for customs organizations since the customs administration is an organization with a high risk of corruption, these customs which are free from corruption, will be more efficient and effective in customs control, and provide a more facilitated and secured business environment to traders. Customs, as a public administration department, his appeal procedures will give traders, agents, and all supply chain parties more reliability, which will also help customs to improve their control performance since their power is under supervision.

## 5. Conclusions

In conclusion, through the analysis of relevant data, this paper finds that PCA can sufficiently reflect the competitiveness of 21 customs authorities, as well as the efforts and achievements of customs reforms in recent years. The directions of WCO instruments and customs' strategy in recent years are proven scientific and effective whilst analyzing the gaps between the benchmark results and putting forward the countermeasures for the existing gaps and deficiencies amongst different national customs. Through this study, we found that the Netherlands, German, and USA customs are the top three customs in terms of competitiveness and found the reason behind these ranks. Logistics performance, capacity building, integrity, and appeal procedures are the key factors to customs control. The customs of these countries, which are good in the areas, achieved ideal rank.

In the future, various reform measures should be implemented to enhance the international competitiveness of customs and spread the experience of best practices better internationally. To enhance the international competitiveness of customs supervision, the following areas, including the supervision of customs clearance, taxation management, and capacity building, may need to be reinforced in the future.

Firstly, customs should continue to streamline the import and export supervision documents, compress the customs clearance time and reduce the compliance cost. Whilst continuously optimizing the business environment, scientific business environment evaluation index systems should be built to reflect the achievements of customs clearance reforms more objectively. In addition, strengthening customs supervision, improving the quality of the basic logistics data ledger, and further making use of logistics data in the prevention and control of security risks at the access and exit stages are important. Furthermore, making plans for the development, transformation, and upgrading of the single window in advance is necessary, especially in combination with the blockchain, to integrate customs supervision with international trade, supply chain, and international finance deeply.

Secondly, customs should focus on the national macro-economic policies and overall development strategies; strengthen taxation investigations; actively participate in the formulation of taxation policies; and establish taxation investigations to serve macro decision-making, economic development, and business environment optimization. Customs also need to implement national tax policies, such as tariff adjustments and tax reductions, to help enterprises to recover from the disruptions of COVID-19.

Thirdly, we found that with the implementation of the capacity building supported by the WCO, the gaps in customs' competitiveness between developed and developing countries have not been huge, suggesting that the WCO has performed a great job in the

past years. However, the need to go forward is still growing to carry out WCO missions, to put the vision and values into practice, and to share the best practices in the area of customs competitiveness with one another. Customs administrations should actively participate in capacity building and anti-corruption governmental cooperation initiated by the WCO, which are also the focus and trends of WCO development in the future. Meanwhile, customs administrations should improve the ability and quality of customs officers as soon as possible and strengthen the cultivation of future talents. Customs colleges and research institutions should continue to exert efforts in personnel training and technological reform to improve the 'productivity and efficiency' of customs officers' supervision effectively.

To the best of the authors' knowledge, most of the previous studies mainly focused on specific commodities and regions that induce a lack of generalization in the research. Also, recent studies mainly concentrated on the adoption of Industry 4.0 (e.g., blockchain, artificial intelligence, e-commerce, machine learning, and Internet of Things), which overlooked the policy perspectives. Based on a series of research results, we may further provide constructive guidelines to create a customs supervision framework under various cultural and geographical settings effectively. To a certain extent, customs supervision is a driving force to boost national competitiveness rankings. It may help build the future of global competitiveness benchmarking. Besides, the applicable tool is a novelty and could be the basis for further research in the same and other areas of industries. Hence, this research may foster knowledge mobilization between industrial practitioners or geographies and construct a suitable framework for possible collaborations between them.

This paper may fall into pitfalls that we may propose in the future research direction. Firstly, this study excluded emerging regions, such as Greater Bay Area and ASEAN countries. These emerging regions will become key players in economic growth after the COVID-19 pandemic. Secondly, this study only relied on secondary data from published reports. Hence, to supplement and validate the research findings, we may conduct semi-structured in-depth interviews with relevant stakeholders, industrial practitioners, policymakers, and researchers. The mixed research method can offset the limitations of qualitative and quantitative research approaches. Thirdly, maritime transport resilience is an 'urgent topic' in response to the COVID-19 pandemic. Nevertheless, the integration of customs supervision into maritime transport resilience is under-researched.

**Author Contributions:** Conceptualization, R.H. and R.W.; methodology, R.H. and R.W.; formal analysis, R.H. and R.W.; investigation, Y.-Y.L.; resources, R.H.; data curation, R.H.; writing—original draft preparation, R.H. and R.W.; writing—review and editing, Y.-Y.L.; supervision, R.H.; project administration, R.H. All authors have read and agreed to the published version of the manuscript.

**Funding:** This research was fully funded by General Administration of Customs P.R. China. The Project name is Research on the Customs Supervision of China-Euro Railway Express from the Perspective of International Supply Chain (Funding No. 2021HK248).

**Data Availability Statement:** Not applicable.

**Conflicts of Interest:** The authors declare no conflict of interest.

## Appendix A

**Table A1.** Data Resources and Index Explanation.

| Data Resource | Burden of Customs Procedure | Logistics Performance | Average Time | Customs Duties in Tax Revenue (%) | OECD-E-Fee and Charges | | WCO Instruments | Customs Officers Productivities |
|---|---|---|---|---|---|---|---|---|
| Index | Customs Clearance and Procedures 1 | Customs Clearance and Procedures 2 | Customs Clearance and Procedures 3 | Duties 1 | Duties 2 | | Guarantee (Security) | Customs Control |
| Explanation | Burden of Customs Procedure measures business executives' perceptions of their country's efficiency of customs procedures. Respondents evaluated the efficiency of customs procedures in their country. The lowest score (1) rates the customs procedure as extremely inefficient, and the highest score (7) as extremely efficient. | The logistics Performance Index overall score reflects perceptions of a country's logistics based on the efficiency of the customs clearance process, quality of trade- and transport-related infrastructure, ease of arranging competitively priced shipments, quality of logistics services, ability to track and trace consignments, and frequency with which shipments reach the consignee within the scheduled time. Respondents evaluate eight markets on six core dimensions on a scale from 1 (worst) to 5 (best). | Documentary compliance captures the time and cost associated with compliance with the documentary requirements of all government agencies of the origin economy, the destination economy and any transit economies. | Customs and other import duties are all levies collected on goods that are entering the country or services delivered by non-residents to residents. They include levies imposed for revenue or protection purposes and determined on a specific or ad valorem basis as long as they are restricted to imported goods or services. | Information on fees | | In the WCO report, there is a total of 3 WCO instruments for customs security. They are RKC, SAFE and HS. If the customs implement three all then get full marks, otherwise get in ratio accordingly. In the WCO General Annex, it is recorded as a guarantee but in the RKC/MC it is revised into security.so we use the security to measure the index | Productivity is equal to Import & Export volume dividing Customs officers' number |
| Data Resource | OECD-G-Automation | OECD-B-Involvement of The Trade Community | OECD-J-Cooperation | OECD-I-Cooperation | OECD-A-Information Availability | OECD C-Advance Ruling | CPI Value | OECD-D-Appeal Procedure |
| Index | Information and communication technology | Customs with 3rd parties 1 | Customs with 3rd parties 2 | Customs with 3rd parties 3 | Information, decision and ruling 1 | Information, decision and ruling 2 | Customs Affairs Complaints & Appeals 1 | Customs Affairs Complaints & Appeals 2 |

**Table A1.** *Cont.*

| Data Resource | Burden of Customs Procedure | Logistics Performance | Average Time | Customs Duties in Tax Revenue (%) | OECD-E-Fee and Charges | | WCO Instruments | Customs Officers Productivities |
|---|---|---|---|---|---|---|---|---|
| Index | Customs Clearance and Procedures 1 | Customs Clearance and Procedures 2 | Customs Clearance and Procedures 3 | Duties 1 | Duties 2 | | Guarantee (Security) | Customs Control |
| Explanation | Electronic import declarations; Electronic export declarations; Procedures for electronic processingElectronic pre-arrival processing; Electronic payment; Processing system-electronic paymentAutomated risk management; Single window IT; Electronic Data Interchange; Automated processing-goods release; Digital certificates; Full-time automated processing; ITC quality | Public consultations; Notice and comment procedures; Consultations guidelines; Targeted Stakeholders; Consultations frequency; Drafts Publication; Public comments; Policy objectives communication | Cross-border coordination; | Internal coordination | Customs Website; Online feedback; Rate of duties information; Enquiry points; Enquiry points operating hours; Enquiry points timeliness; Import/Export procedure information; Accessible documentation; Advance Publication; Advance Publication-time; Agreements publication; Appeal Procedures information; Customs Classification Examples; Advance rulings information; Breaches formalities; Application legislation; Judicial decision; Professional users site; User manuals; Website user friendless; Policy making transparency | Advance rulings; ARs tariff number; ARs origin number; ARs total number; ARs validity; ARs issuance time publication; ARs issuance time; ARs within issuance time; ARs information; ARs review request; ARs refusal motivation | The Corruption Perceptions Index (CPI) aggregates data from various sources on the perception of the level of corruption in the public sector by country experts and business representatives. As part of the publication of the CPI, a standard deviation and a confidence interval for the CPI value are specified, which depict the variance of the assessments of a country/area within the framework of the available data sources. | Appeal procedural rules; Judicial appeals; Appeal lodging time; Appeal delays; Appeal information motives; Appeals introduced by customs; Appeals introduced by traders; Administrative appeals-number; Judicial appeals number; Judicial appeal time limit; Appeal time limit decision; Legal framework efficiency; Judicial independence |

**Table A2.** Data from 21 Selected Countries.

| RKC-Evaluation Elements<br><br>Country/Index | Customs Clearance and Procedures 1<br><br>Burden of Customs Procedure | Customs Clearance and Procedures 2<br><br>Logistics Performance | Customs Clearance and Procedures 3<br><br>Average Time | Duties 1<br><br>Customs Duties in Tax Revenue (%) | Duties 2<br><br>Fee and Charges | Guarantee Security<br><br>WCO Instruments Implement | Customs Control<br><br>Customs Officers Productivities | Information and Communication Technology<br><br>OECD-G-Automation | Customs with 3rd Parties 1<br><br>OECD-B-Involvement of the Trade Community | Customs with 3rd Parties 2<br><br>OECD-J-Cooperation | Customs with 3rd Parties 3<br><br>OECD-I-Cooperation | Information, Decision and Ruling 1<br><br>OECD-A-Information Availability | Information, Decision and Ruling 2<br><br>OECD C-Advance Ruling | Customs Affairs Complaints & Appeals 1<br><br>CPI Value |
|---|---|---|---|---|---|---|---|---|---|---|---|---|---|---|
| China | 4.600 | 3.610 | 10.700 | 3.200 | 1.923 | 1.000 | 835.5383118 | 1.600 | 1.857 | 0.800 | 1.000 | 1.476 | 2.000 | 41.000 |
| Argentina | 2.800 | 2.890 | 95.500 | 13.800 | 1.769 | 1.000 | 213.0135979 | 1.462 | 1.429 | 1.182 | 1.300 | 1.429 | 1.714 | 45.000 |
| Australia | 5.000 | 3.750 | 5.500 | 3.500 | 1.857 | 1.000 | 1013.981472 | 1.769 | 1.875 | 1.636 | 1.545 | 2.000 | 1.909 | 77.000 |
| Brazil | 3.000 | 2.990 | 18.000 | 8.800 | 1.846 | 0.667 | 1498.804906 | 1.426 | 1.375 | 1.091 | 0.909 | 1.571 | 1.636 | 35.000 |
| Canada | 5.200 | 3.730 | 1.000 | 2.500 | 1.786 | 1.000 | 1560.299571 | 1.615 | 1.875 | 1.727 | 1.545 | 1.857 | 1.636 | 77.000 |
| France | 4.800 | 3.840 | 0.500 | 0.700 | 1.846 | 1.000 | 563.4842435 | 1.615 | 1.750 | 1.727 | 1.545 | 1.810 | 1.909 | 69.000 |
| Germany | 5.300 | 4.200 | 0.750 | 0.000 | 1.857 | 1.000 | 6738.434839 | 1.667 | 1.857 | 1.727 | 1.545 | 1.810 | 1.818 | 80.000 |
| India | 4.600 | 3.180 | 15.750 | 8.100 | 1.692 | 1.000 | 792.742697 | 1.692 | 1.429 | 0.909 | 1.909 | 1.905 | 1.300 | 41.000 |
| Indonesia | 4.200 | 3.150 | 83.750 | 3.000 | 1.538 | 1.000 | 220.5045801 | 1.000 | 1.571 | 0.818 | 0.900 | 1.524 | 1.400 | 40.000 |
| Italy | 4.300 | 3.740 | 0.500 | 0.500 | 1.846 | 1.000 | 2118.58187 | 1.615 | 1.375 | 1.636 | 1.364 | 1.650 | 1.909 | 53.000 |
| Japan | 5.000 | 4.030 | 2.900 | 1.700 | 1.786 | 1.000 | 1462.650112 | 1.833 | 1.750 | 1.909 | 1.273 | 1.800 | 2.000 | 73.000 |
| Mexico | 4.100 | 3.050 | 12.800 | 2.000 | 1.769 | 0.667 | 1211.553917 | 1.923 | 1.625 | 1.545 | 1.455 | 1.238 | 1.500 | 29.000 |
| Saudi Arabia | 4.800 | 3.010 | 21.500 | 6.135 | 1.846 | 1.000 | 267.3550893 | 1.615 | 1.625 | 0.909 | 0.909 | 1.429 | 1.667 | 53.000 |
| South Africa | 4.200 | 3.380 | 52.000 | 4.000 | 1.846 | 1.000 | 4170.961457 | 1.900 | 1.571 | 1.200 | 0.909 | 1.619 | 1.667 | 59.000 |
| Turkey | 3.900 | 3.150 | 3.000 | 1.400 | 1.615 | 1.000 | 404.430708 | 1.667 | 1.750 | 0.909 | 1.700 | 1.476 | 1.364 | 44.000 |
| British | 5.500 | 3.990 | 2.850 | 0.400 | 1.846 | 1.000 | 271.5389992 | 1.917 | 1.875 | 2.000 | 1.636 | 1.810 | 1.909 | 39.000 |
| USA | 5.600 | 3.890 | 4.500 | 2.100 | 1.929 | 1.000 | 909.1293977 | 1.923 | 1.875 | 1.455 | 1.900 | 1.952 | 1.857 | 82.000 |
| Korea | 4.600 | 3.610 | 1.000 | 2.500 | 1.846 | 1.000 | 6520.442707 | 2.000 | 2.000 | 1.899 | 1.909 | 1.905 | 2.000 | 69.000 |
| Netherlands | 5.800 | 4.020 | 0.500 | 1.700 | 1.929 | 1.000 | 1818.173303 | 2.000 | 1.875 | 1.909 | 1.545 | 1.952 | 1.909 | 82.000 |
| New Zealand | 5.800 | 3.880 | 2.000 | 3.400 | 1.929 | 1.000 | 1429.211877 | 2.000 | 1.750 | 1.727 | 1.727 | 1.810 | 1.455 | 87.000 |
| Singapore | 6.300 | 4.000 | 2.500 | 0.000 | 1.923 | 0.667 | 8573.165489 | 2.000 | 2.000 | 0.909 | 1.500 | 1.905 | 2.000 | 85.000 |

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
