# Peer review of "Evaluation of Customs Supervision Competitiveness Using Principal Component Analysis"

_sustainability, doi:10.3390/su15031833_

Round 1
Reviewer 1 Report
The paper evaluates Customs status in 21 countries. The applicable tool is interesting and could be the basis for further research in both the same and other areas of industries. However, I would strongly suggest to follow the standard structure of a paper with a separate literature review section that will also include relevant research except the referenced reports. A literature review will most probably also improve the paper as whole. For instance the identification of literature will identify areas of improvement of the power in the results section because the methodology only ranks Customs regimes but cannot identify root causes. The literature might help in this area as well.
I hope the above are useful.
All the best to the Authors
Author Response
Response: Thanks for the reviewer’s comments. We have moved some relevant content from the introduction to the literature review section. Also, the literature review is further improved in Section 2. The paper also did the editing process.
Reviewer 2 Report
Dear Authors,
The topic and the background of the study are quite interesting. However, there are certain parts that need clarification and actions from the authors, in order to improve the quality of the paper and to avoid ethical concerns.
Please find the attachment to see my comments for improvement.
Thanks.

Author Response
To me, the title of this paper is quite weak as it did not highlight what the study is all about. My suggestion to the authors is to revise the title accordingly by highlighting the key components of the study. For example; The Evaluation of Custom Supervision Competitiveness Using the Principal Component Analysis (PCA) Method
Response: Thanks for the reviewer’s comments. We have revised the paper title accordingly.
The contents of the abstract were also clear and concise.
Response: Thanks for the reviewer’s positive comments.
However, the contents of the abstract were supposed to highlight the findings of the study rather than highlighting the other studies. Please response to this statement “Based on the data from the World Economic Forum, World Bank, OECD, WCO Annual Report, and Transparency International, it can be found that the Dutch customs have the relatively best performance in the range of comprehensive competitiveness, and customs authorities in Germany, New Zealand, the United Kingdom, the United States, Mexico, Australia, Netherlands, and Singapore also have relatively best performance under different indexes respectively.” Is this based on your own analysis, or the output of other studies? If it is from other studies, please revise accordingly.
Response: Thanks for the reviewer’s comments. From the data collected from World Bank, OECD, and WCO reports, we use PCA to conduct the analysis and then draw above mentioned conclusion. Not from other studies.
In addition, the abstract of this paper was not meeting the format set by the Sustainability journal, which is maximum 200 words only (See Instruction for Authors). The abstract of the paper was found to be 224 words in total. Hence, the authors are recommended to rephrase or paraphrase the contents in the abstract to meet the format given by the journal without affecting the clarity of the information.
Response: Thanks for the reviewer’s comments. We have further reduced the number of wordings to meet the requirements of journal abstract requirements.
The Introduction part was too long and ineffective to attract readers to read this paper. Several parts should be restructured and moved to Literature Review part to be more effective.
Response: Thanks for the reviewer’s comments. We have moved some relevant content from the introduction to the literature review section.
Page 2, 2nd Paragraph, line 54-58 – “From the point of WCO Research, performance measurement has been discussed at the international grand level and also in the context of a single country. At the international level, the paper concerning an overview of performance measurement in customs administrations considers four broad approaches, Customs Data Mining, Service Charters, Perception Indexes, and Monitoring Mechanisms.” (No citation provided for this statement)
Response: Thanks for the reviewer comments. We have already provided the in-text citation. The reference is “Ireland, R.; Cantens, T.; Yasui, T. An overview of performance measurement in customs administrations. WCO Research Paper 2011, 13”.
Page 2, 2nd Paragraph, line 61-64 – “…, another WCO research paper carries out a comparative study 61 on cross-border trade indicators. It mainly compares the differences between the Ease of 62 Doing Business Index, Logistics Performance Index, and Trade Facilitation Index etc., and 63 emphasizes the importance of the WCO Time Release Study [5].” It supposes to be past tense, since the study has been conducted in past.
Response: Thanks for the reviewer’s comments. We have made the correct past tense in the above sentences.
Page 2, 2nd Paragraph, line 69-70 – The paper analyzes these results and concludes that the three selection methods are complementary for the detection and deterrence of emerging and evolving risks [6].” What results do you meant? You should indicate which research or results that you are referring for this statement.
Response: Thanks for the reviewer comments. When we discuss customs risk management, it is very important and useful to figure out the selection methods in customs risk detection.
Many statements have no proper citation provided. Please double check and provided related citations accordingly. Otherwise, it will raise ethical concerns on this paper.
For examples,
- Page 2, Paragraph 4 - Principal Component Analysis (PCA), Data Envelopment Analysis (DEA), and the combination of Panel Data Regression and Growth Curve Analysis have been used in terms of evaluation of the organizational performance of different customs services.
Response: Thanks for the reviewer’s comments. The statement mentioned above is a conclusive description made by the authors at the beginning of the paragraph, summarizing the content of the whole paragraph without reference to any existing literature, which refers to PCA, DEA, and the combination of Panel Data Regression and Growth Curve Analysis respectively. The relevant citations are located within the paragraph and after this conclusive description.
- Page 2, Paragraph 4 - Based on data from the WTO, WCO, and WB, PCA has been used to assess the competitiveness of different customs authorities from 29 countries under the framework of international trade.
Response: Thanks for the reviewer’s comments. Actually, the statement mentioned above has its corresponding citation, which comes after several sentences. Torres & Chávez (2015) is the citation of this statement, we take a few sentences to describe the content of this paper. As you can see from the screenshot below, the citation has fully correspondence with the content of the statement.
- Page 2, Paragraph 4 - During the 15th WCO PICARD Conference in 2020, organizational performance measurement was among one the heated topics with presenters using different methods for scientific evaluation. DEA has been used to assess the relative efficiency of the implementation of WCO Policies and Guidelines by the customs and commercial community on the basis of two models: Constant Return to Scale (CRS) and Variable Return to Scale (VRS).
Response: Thanks for the reviewer’s comments. The statement mentioned above needs to be divided into two parts for an explanation.
First, “During the 15th WCO PICARD Conference in 2020, organizational performance measurement was among one the heated topics with presenters using different methods for scientific evaluation”, this part is a conclusive description made by the authors, which summarizes the relevant following contents without reference to any existing literature.
Second, “DEA has been used to assess the relative efficiency of the implementation of WCO Policies and Guidelines by the customs and commercial community on the basis of two models: Constant Return to Scale (CRS) and Variable Return to Scale (VRS)”, this part has its corresponding citation, which comes after several sentences. “(Rodríguez Lozano, 2020)” is the citation of this statement, we take a few sentences to describe the content of this paper. The citation has full correspondence with the content of the statement.
Literature Review Part is recommended to be created to improve the structure of the paper and named as Section 2.
Response: Thanks for the reviewer’s comments. We have created another new Section
2 for the literature review part.
Section 2 is recommended to be renamed to Section 3: Materials and Methods / Methodology.
Response: Thanks for the reviewer’s comments. We have revised from Section 2 to Section 3 and renamed the name of the section accordingly.
Some statements need proper citation to be provided. Please double check and provided related citations accordingly. Otherwise, it will raise ethical concerns on this paper.
For examples,
- Section 2, 2.1, Paragraph 1 – “The Revised Kyoto Convention is the only international legal document that comprehensively regulates the standards of customs systems and practices in the world today. It is an important basis for the WCO to promote the coordinated development of customs systems and practices in various countries and regions around the world, and also an important reference standard for each country and region to formulate its own customs systems in order to strengthen supervision, promote and facilitate the development of trade. Judging from the revision work of the Revised Kyoto Convention launched in 2019, customs supervision of all member countries still follows the above-mentioned framework. Therefore, the evaluation index system of international competitiveness for customs from nearly all countries in the world can almost fully refer to the Revised Kyoto Convention.”
Response: Thanks for the reviewer’s comment. The statement mentioned above is not derived from any existing literature, but an original conclusive description made by the author. One of the three authors has personally joined the process of a “comprehensive review of the Revised Kyoto Convention” launched by the World Customs Organization (WCO) in 2019 as a representative from academia. Besides, as a practitioner in customs affairs, she also has rich working experience in China Customs for many years previously. Therefore, the conclusive deduction mentioned above is well-underpinned.
Section 2, 2.1, Paragraph 2 – “According to the Revised Kyoto Convention, technical standards are mainly set up in the areas of customs clearance procedures, duties, guarantee (security), customs supervision, information & communication technology, customs relationship with third parties, information & decisions and rulings provided by customs, and customs affairs complaints and appeals etc. This provides an ideal basis for the evaluation of customs competitiveness internationally.”
Response: Thanks for the reviewer’s comment. The statement mentioned above is not derived from any existing literature, but an original conclusive description made by the author. One of the three authors has personally joined the process of a “comprehensive review of the Revised Kyoto Convention” launched by the World Customs Organization (WCO) in 2019 as a representative from academia. Besides, as a practitioner in customs affairs, she also has rich working experience in China Customs for many years previously. Therefore, the conclusive deduction mentioned above is well-underpinned.
Section 3, is recommended to be renamed to Section 4: Principal Component Analysis of International Competitiveness of Customs Supervision
Response: Thanks for the reviewer’s comments. We have changed from Section 3 to Section 4 accordingly. The Section 4 title also changed to Results and Discussion will be clearer.
The contents of Section 3 are clear and appropriately highlighted.
Response: Thanks for the reviewer’s positive comments.
Section 4, is recommended to be renamed to Section 5: Conclusions
Response: Thanks for the reviewer’s comments. We have renamed Section 4 into Section 5 as Conclusions.
Section 5 is combined together with Section 4.
Response: Thanks for the reviewer’s comments. We have combined Sections 4 and 5 together.
The contents of Section 4 and 5 are clear and appropriately highlighted.
Response: Thanks for the reviewer’s positive comments.
The contents of Section 4 and 5 are clear and appropriately highlighted.
Response: Thanks for the reviewer’s positive comments.
The revised manuscript also did the editing.
Reviewer 3 Report
The topic is interesting and offers a useful approach to assess custom supervision. The adoption of the principal component analysis approach is also one of the strengths of the article. However, I don’t feel that the paper is ready for publication, mainly because of the writing style and how the article is organised. Please see my specific comments below.
1. I don’t see the link between the topic of the research and the issue of sustainability. This has to be made more explicit.
2. I think the section on introduction is a bit dry, focused mainly on technical aspects of the custom performance and relevant indicators. But it is difficult to understand from this reading what the authors want to achieve. I suggest adding a non-technical initial paragraph that explains the objective of the research, what the research problem is, what the research gap is, and how this section is organised. This will help the readers to understand from the beginning the direction and focus of the research.
3. The heading of subsection 2.1. says “Evaluation Index Elements of International Competitiveness of Customs Supervision and 180 Data Sources”. I don’t really see any evaluation of the indexes. What the authors do, is to mention existing indexes. I suggest changing the name of this subsection, and also add a table with the information of the selected indexes and their meaning.
4. The title of Section 2 is misleading. After reading some paragraph I realised that this section is actually materials and methods. This has to be amended. This section also has results from the principal component analysis. This should be placed in the section on results and discussion.
5. The sentence at the end of Table 2 is misleading. It says Source: WCO, World Bank, OECD, Transparency International, and SPSSAU. But this is incorrect. I agree that the database was obtained from these sources. But the numbers shown in the table come from the PCA, which was carried by the authors, not by these organisations.
6. The way in which results are presented is very robotic. I suggest using less subheadings and more interesting narratives. For this, the authors should explain the implications of the findings.
7. I am glad to see that the authors propose ways to enhancing the competitiveness of customs supervision. But again, the way in which this is presented is very robotic and dry. It looks like a sort of dissertation more than an academic article. Please use less subheadings and introduce more interesting narratives with more implications for the actors who participate in custom activities.
8. Conclusions are very poor. Please extend it by making sure that the following points are covered: brief description of the research problem and the novelty of the study; brief description of the results; implication of the results; limitations, and proposals for future research.
Author Response
I don’t see the link between the topic of the research and the issue of sustainability. This has to be made more explicit.
Response: Thanks for the reviewer’s comments. We have provided a link between the research and the issue of sustainability in the introduction section.
- I think the section on introduction is a bit dry, focused mainly on technical aspects of the custom performance and relevant indicators. But it is difficult to understand from this reading what the authors want to achieve. I suggest adding a non-technical initial paragraph that explains the objective of the research, what the research problem is, what the research gap is, and how this section is organized. This will help the readers to understand from the beginning the direction and focus of the research.
Response: Thanks for the reviewer’s comments. We have added the sentences to help readers to understand from the beginning the directions and focus of the research in the introduction section.
- The heading of subsection 2.1. says “Evaluation Index Elements of International Competitiveness of Customs Supervision and 180 Data Sources”. I don’t really see any evaluation of the indexes. What the authors do, is to mention existing indexes. I suggest changing the name of this subsection, and also add a table with the information of the selected indexes and their meaning.
Response: Thanks for the reviewer’s comments. Please refer to Annex 1a, Annex 1b, and Annex 2.
- The title of Section 2 is misleading. After reading some paragraph I realised that this section is actually materials and methods. This has to be amended. This section also has results from the principal component analysis. This should be placed in the section on results and discussion.
Response: Thanks for the reviewer’s comments. We have added Section 2 as a literature review. Section 3 is now named materials and methods. Section 4 is named Results and Discussion and Section 5 is named Conclusions
- The sentence at the end of Table 2 is misleading. It says Source: WCO, World Bank, OECD, Transparency International, and SPSSAU. But this is incorrect. I agree that the database was obtained from these sources. But the numbers shown in the table come from the PCA, which was carried by the authors, not by these organisations.
Response: Thanks for the reviewer’s comments. We have removed the source of references in Table 2.
- The way in which results are presented is very robotic. I suggest using less subheadings and more interesting narratives. For this, the authors should explain the implications of the findings.
Response: Thanks for the reviewer’s comments. We have added sentences to explain the implication of the findings.
- I am glad to see that the authors propose ways to enhancing the competitiveness of customs supervision. But again, the way in which this is presented is very robotic and dry. It looks like a sort of dissertation more than an academic article. Please use less subheadings and introduce more interesting narratives with more implications for the actors who participate in custom activities.
Response: Thanks for the reviewer’s comments. We have added sentences to explain the implication of the findings. Also, we have used less subheadings for the research result implications.
- Conclusions are very poor. Please extend it by making sure that the following points are covered: brief description of the research problem and the novelty of the study; brief description of the results; implication of the results; limitations, and proposals for future research.
Response: Thanks for the reviewer’s comments. We have added it in the conclusion section.
The revised manuscript also did the editing.
Round 2
Reviewer 1 Report
Thank you for taking the time to address my comments. All the best.